# Cardiovascular Complications in Community-Acquired Pneumonia

**DOI:** 10.3390/microorganisms10112177

**Published:** 2022-11-02

**Authors:** Antonio Desai, Stefano Aliberti, Francesco Amati, Anna Stainer, Antonio Voza

**Affiliations:** 1IRCCS Humanitas Research Hospital, Emergency Department, Via Manzoni 56, Rozzano, 20089 Milan, Italy; 2Department of Biomedical Sciences, Humanitas University, Via Rita Levi Montalcini 4, Pieve Emanuele, 20072 Milan, Italy; 3IRCCS Humanitas Research Hospital, Respiratory Unit, Via Manzoni 56, Rozzano, 20089 Milan, Italy

**Keywords:** acute myocardial infarction, arrhythmia, cardiovascular complications, community-acquired pneumonia, heart failure, stroke

## Abstract

Community-acquired pneumonia (CAP) is accountable for high mortality in both pediatric and adult populations worldwide, about one-third of hospitalized patients pass away within a year of being discharged from the facility. The high mortality and morbidity rates are closely related to cardiovascular complications that are consequent or concomitant to the acute episode of pneumonia. An updated perspective on the major pathophysiological mechanisms, prevalence, risk factors, outcomes, and relevant treatments of cardiovascular events in CAP patients is provided in the current study. It is possible to evaluate the pathophysiology of cardiac disease in this population based on plaque-related events, such as acute myocardial infarction, or events unrelated to plaque, such as arrhythmias and heart failure. With an absolute rate of cardiovascular problems ranging broadly from 10% to 30%, CAP raises the risk of both plaque-related and plaque-unrelated events. Both in- and out-patients may experience these issues at admission, throughout hospitalization, or even up to a year following discharge. At long-term follow-up, cardiac events account for more than 30% of deaths in CAP patients, making them a significant cause of mortality. If patients at risk for cardiac events are stratified, diagnostic tools, monitoring, and preventive measures may be applied to these patients. A prospective evaluation of cardioprotective treatments is urgently required from a research point of view.

## 1. Introduction

Community-acquired pneumonia (CAP) is the most common and deadly respiratory infection throughout western countries, affecting both immunosuppressed and healthy patients in the pediatric and adult populations [1]. CAP mostly consists of an acute bacterial or viral infection of the lung parenchyma clinically characterized by fever, cough, and respiratory failure [2]. Despite recent advances in diagnostics, therapeutic strategies, and the implementation of vaccines, the morbidity and mortality for CAP persisted very high. Mortality for CAP is 3%, 23%, and 31%, at hospitalization, 30 days, and 1 year, respectively [3,4,5,6].

The poor prognosis is closely related to concomitant complications, such as respiratory failure, sepsis, and cardiovascular events which strongly worsen patients’ outcomes [7,8,9]. Notably, cardiological events, such as acute coronary syndromes, account for 30% of the long-term mortality [10,11].

The correlation between CAP and heart disease depends on several factors. Pneumonia may lead to a systemic inflammatory response inducing severe hypoperfusion and multiorgan failure [12,13]. The strong systemic inflammatory reaction with a quick boost of pro-inflammatory cytokines, C-reactive proteins (CRP), interleukins (IL), and tumor necrosis factor (TNF), induces the so-called “systemic inflammatory syndrome” which evolves in severe hypoperfusion and multiorgan failure, comprising sepsis and/or septic shock [14,15,16,17,18]. The ventilation-perfusion mismatch and intrapulmonary shunt result in hypoxemia, a temporary rising in serum concentrations of endothelin-1, a vasoconstrictor, and an increase in the coagulation cascade [19,20]. The pro-thrombotic *status* consequent to CAP, mediated by an increase in thrombin–antithrombin complex, plasminogen activator inhibitor, and D-dimer concentrations, is related to poor outcomes in CAP patients [20,21,22,23]. The pathophysiology of CAP-related cardiovascular complications results from the process of atherosclerosis leading to endothelial injury and myocardial ischemia or dysfunction [24,25].

The aim of this article is to summarize the main mechanisms and inflammatory pathways liable for the occurrence of cardiovascular complications. The predictive factors, clinical implications, and future perspectives in the therapeutic and diagnostic fields will be also discussed.

We searched Medline from inception to 1 August 2021 for articles in English evaluating the incidence of subsequent cardiovascular events during and after a CAP episode (community-acquired pneumonia” AND (“complications” OR “acute coronary syndromes” OR “heart failure” OR “arrhythmia” OR “atrial fibrillation” OR “stroke” OR “acute myocardial infarction”). We also retrieved previously published reviews on the subject and scanned their references for any additional missed publications.

## 2. Cardiovascular Complication in Community-Acquired Pneumonia

### 2.1. Plaque-Related Cardiovascular Complications: Acute Myocardial Infarction

Atherosclerosis and endothelial dysfunction are responsible for plaque rupture, thrombus creation, and mismatch between arterial perfusion and myocardial oxygen request, and are the basis of different vascular pathologies [26]. Indeed, a myocardial infarction occurs in those with atherosclerotic plaque rupture and thrombosis (type I), or in case of myocardial oxygen supply and demand imbalance in the context of an acute illness (type II). After plaque disruption, local cardiac ischemia may occur. The rupture of a fibrous cap overlying a previously stable plaque is the *primum movens* of cardiovascular events, such as acute myocardial infarction (AMI) and unstable angina. Plaque rupture exposes the underlying lipid-rich core which is highly pro-thrombotic with the subsequent formation of a thrombus in situ and the occlusion of the vessel [27]. Platelet activation and aggregation on the surface of a ruptured plaque may be stimulated by pneumonia either directly through pro-inflammatory cytokines or by bacterial products [28,29]. All these processes, comprising the plaque rupture, in situ thrombus formation, and the balance of arterial perfusion versus myocardial oxygen demand may be triggered or worsened during an episode of pneumonia and, thus, lead to the occurrence of plaque-related cardiovascular events.

Since the early part of the 20th century, pneumonia and plaque-related events have been linked [30,31,32]. The discovery of a considerable temporal increase in the incidence of acute coronary syndrome shortly after the onset of respiratory infections was one of the earliest indicators of the link between CAP and AMI [33,34].

There was an elevated incidence of this occurrence following respiratory infections for up to 90 days, according to three significant primary care studies involving more than 33,000 patients with the first episode of AMI [33,34,35]. Furthermore, one of the most interesting results of these studies was the minimal or no association between urinary tract infection and AMI, suggesting that the increased risk for cardiovascular events may be specific to respiratory infections. Among longitudinal cohort studies, a particular interest has been focused on CAP caused by S. pneumoniae. Wang and coworkers found an increased risk of the acute coronary syndrome in patients with pneumococcal pneumonia in the long-term follow-up and that pneumococcal pneumonia patients have a higher relative risk than patients without pneumococcal pneumonia [36].

Furthermore, those who had CAP brought on by *S. pneumoniae* or *H. influenzae* showed a greater risk of cardiac events in the 15 days following hospitalization. Additionally, as compared to the risk 1 year prior to the episode of CAP, the risk of AMI rises by more than 40 times [11,36]. According to a meta-analysis, 5% of individuals with pneumonia had either an AMI or unstable angina (range 1–11%) [10].

Even after reaching clinical stability, a persisting subclinical inflammation can be detected in patients with CAP at the time of hospital discharge. Aliberti et al. demonstrated in a cohort of 500 patients hospitalized with CAP that the most common etiology for clinical failure related to pneumonia after severe sepsis was the occurrence of AMI [24]. Higher levels of IL-6 and IL-10 at hospital discharge are linked to a higher risk of death at one-year follow-up, according to research by Yende and colleagues [37]. According to these hypotheses, plaque-related events that occur during the acute phase of the infection and for a longer duration later may be the cause of the elevated systemic inflammatory status in patients with respiratory infections.

### 2.2. Plaque-Unrelated Cardiovascular Complications: Arrhythmias and Heart Failure

The acute or persistent inflammatory reaction and hypoxemia happening during CAP are responsible for diffuse organ dysfunction, arrhythmias, and heart failure. Many mechanisms can be used to explain how plaque-unrelated heart failure develops. Because of the ventilation/perfusion mismatch or shunt during the episode of pneumonia, the harmful inflammatory effects on the heart induce myocardial dysfunction, reduced myocardial contractility, greater myocardial oxygen demand, and lower myocardial oxygen supply, leading to heart failure [10,38,39]. Moreover, acute renal impairment is frequent in hospitalized patients with CAP and plays a pivotal role in heart failure by several mechanisms such as alteration in the renin-angiotensin-aldosterone system (RAAS), dysregulated pressure-sensing baroreceptors and cellular signaling, abnormal sympathetic nervous system mechanisms [40,41].

Cardiac arrhythmias, such as atrial tachyarrhythmias and atrial fibrillation, as well as cardiac conduction anomalies concurrent to CAP, were identified in the early 20th century and confirmed thereafter [31,42]. In a study of more than 800,000 atrial fibrillation patients treated in hospitals, pneumonia was shown to be the second most common main diagnosis in patients 65 years of age or older (7%), trailing only congestive heart failure (13%) and coming before acute myocardial infarction (6%) [43]. Recent observational studies’ data revealed that between 1% to 11% percent of hospitalized CAP patients experienced a new or worsening pre-existing arrhythmia [10,11,38,44]. According to a recent meta-analysis, 14% of CAP patients who were hospitalized had heart failure (range 7–33%) [10]. The authors reported the incidence of heart failure as a consequence of CAP is more common in the elderly and in patients with pre-existing coronary artery disease [10]. Other observational studies found that CAP hospitalized patients had a prevalence of heart failure ranging from 3% to 18% [10]. Finally, epidemiological evidence points to pre-existing heart failure as a risk factor for pneumonia development, raising the possibility that the cause-and-effect relationship between these two occurrences may be bidirectional [45,46].

## 3. Impact and Predictors of Cardiovascular Complications on CAP Patients

Shreds of literature evidence reported increased short-term mortality in patients with concurrent CAP and cardiac events, ranging from 15% to 36% and with an odds ratio (OR) from 1.6 to 3.9 [38,47,48,49]. Cardiovascular events are also related to 90-day mortality with an OR of 1.93 for AMI and 2.39 for atrial fibrillation [50]. All these data confirmed previous findings by Mortensen et al. who reported that, after neurological conditions and malignancy, cardiac ischemia was the third most common underlying cause of death 90 days after an episode of pneumonia [3].

During hospitalization, cardiovascular complications are responsible for the major causes of long-term morbidity and mortality [7,37,51,52]. Recently, Bruns et al. reported an increased risk of death at 1, 5, and 7 years following an episode of CAP compared with the general Dutch population [8]. Overall, 16% of deaths in this cohort were caused by cardiovascular events. Elderly patients were observed in a population-based study from Finland for a mean of 9.2 years, and the researchers found that they had a considerably higher risk of long-term death (relative risk: 2.4) [53].

Given the effect cardiovascular events have on CAP patients’ outcomes, clinicians must focus on managing and preventing cardiovascular events from occurring. Recently, some studies focused on the predictors for the occurrence of cardiovascular events in patients with pneumonia, including demographics (age and nursing home residency), comorbidities (COPD, diabetes, prior AMI, hyperlipidemia, history of cardiac arrhythmias, coronary artery diseases, hypertension, congestive heart failure, chronic kidney diseases, hypoalbuminemia), the severity of the disease on admission (pneumonia severity index -PSI-, septic shock, multilobar involvement, and pleural effusion, tachycardia and tachypnea, arterial pH < 7.35, blood urea nitrogen > 30, sodium < 130, hematocrit < 30), and microbiology (*S. aureus* and *S. pneumoniae*) [10,48,49]. A prospective study conducted by Corrlas-Medina et al. identified older age, nursing home residence, preexisting cardiovascular disease, and pneumonia severity with cardiovascular events [54].

A high raised level of NT-proBNP has been found to be a strong predictor of early mortality following the hospital admission for CAP and this is independent of existing clinical risk prediction scores [55]. Clinical risk factors are categorized in Table 1.

Identifying the most vulnerable patients who are likely to develop cardiovascular events can be challenging. For both therapeutic and scientific objectives, clinical methods that categorize CAP patients thought prediction scores based on their propensity to experience cardiac problems may be helpful. Using clinical and laboratory data available at the time of hospital admission, recent experiments have attempted to develop a prediction score to stratify the short-term risk of cardiac complications [10,48,56,57].

In 2013, Viasus and colleagues conducted a prospective monocenter study to identify a prediction score that stratifies low-, intermediate- and high-risk groups [10,48]. Corrales-Medina et al. developed and validated a prediction rule for the short-term risk of cardiac complications with more accuracy than the PSI score, using standard clinical and laboratory data. this strategy may be useful to validate a timely risk stratification in clinical and research settings [10]. A major challenge in future CAP research will be the evaluation of cardiovascular biomarkers in addition to risk factors to improve clinical scoring and, thus, better identify patients at risk of cardiovascular events.

## 4. Cardioprotective Interventions

Therapeutic strategies aimed to prevent the occurrence of cardiac complications in patients with CAP may be distinguished according to the causative pathological mechanisms.

In the context of atherosclerosis, the main approaches include the administration of drugs able to stabilize the plaque, namely statins and aspirin. Statins improve outcomes of hospitalized patients with CAP because of the pleiotropic effects, immunomodulation, anti-inflammatory, and anti-oxidative effects, and improvement of endothelial function [58]. Statins might be beneficial for reducing the risk of CAP-related mortality and morbidity, according to a meta-analysis [59]. However, because of observational designs and demographic heterogeneity, the results are very low-quality data. In patients with pneumonia and high cardiovascular risk, anti-inflammatory medication, and methods to prevent platelet activation and aggregation may help lower the incidence of coronary syndromes. A retrospective observational analysis of CAP patients revealed a non-significant mortality rate reduction associated with the preventive use of aspirin. Aspirin significantly lowers the risk for cardiac events (10.6% in the control group vs. 1.1% in the aspirin group; relative risk: 0.103, *p*-value = 0.015) as well as cardiovascular mortality, according to a randomized clinical trial involving 185 CAP patients who were at intermediate to high risk for cardiovascular disease [60].

Protecting cardiomyocytes that have been harmed by acute infections is the primary method of preventing the onset of events unrelated to plaque. The preservation of metabolic function, a decrease in inflammation, and the production of cytotoxic agents are all aspects of protecting cardiac cells. Free radical toxicity, neutrophil infiltration, and matrix breakdown are the treatment targets. Potentially helpful therapies could include the use of antioxidants and free radical scavengers, anti-cytokine antibodies, and control of the activity of the molecules that cause matrix breakdown. An improvement in medical treatment of cardiovascular comorbidities and prevention of cardiac events during an episode of CAP may give the possibility of strongly reducing mortality. However, intervention trials in this field are warranted since most of the data on cardioprotective medications in patients with CAP are coming from observational studies.

## 5. Cardiovascular Events in COVID-19 Patients

A pandemic and an unparalleled worldwide public health disaster, the Coronavirus Disease 2019 (COVID-19) is accompanied by vicariously stressful psychosocial and economic conditions. According to reports, people who are not infected with the COVID-19 virus experience behavioral and emotional disorders as a result of such pressures. The likelihood that these pressures will result in acute cardiovascular events (CVE) in those people is mainly unknown [61].

The most prevalent concomitant condition among COVID-19 patients, cardiovascular disease is directly correlated with the disease’s severity. Acute myocardial damage and myocarditis, heart failure and cardiac arrest, arrhythmia, acute myocardial infarction, cardiogenic shock, Takotsubo cardiomyopathy, and coagulation abnormalities are only a few of the cardiac problems that SARS-CoV-2 infection can directly or indirectly induce [62]. According to Ruan et al., SARS-CoV-2 infection can cause fulminant myocarditis, which can lead to acute cardiac damage in COVID-19 patients, resulting in acute myocardial injury in patients with COVID-19 [63].

Moreover, a viral infection can exacerbate heart failure or even cause acute heart failure. SARS-CoV and MERS infections have been linked to instances of heart failure in the past. A clinical research of 99 cases of COVID-19 from Wuhan recently revealed that 11 (11%) of the patients had passed away, of whom two had no prior history of chronic heart disease but developed heart failure and ultimately passed away from a sudden cardiac arrest [64,65,66,67]. One of the earliest symptoms in certain COVID-19 patients was also mentioned as heart palpitations, which suggests a potential arrhythmia brought on by SARS-CoV-2 infection [68]. Acute myocardial infarction has been noted as a frequent cardiac consequence in individuals with viral pneumonia [69]. Notably, multiple investigations revealed that individuals with severe COVID-19 exhibited raised D-dimer concentrations and enhanced coagulation activity (>1 g/L). The likelihood that an atherosclerotic plaque would rupture and cause an AMI may be increased by local inflammation, the production of procoagulant factors, and hemodynamic abnormalities [70].

Through a variety of methods, including direct injury, downregulation of ACE2, immunological injury, hypoxic injury, and psychological stress, SARS-CoV-2 causes viral infection-related heart damage. As a result, it is critical to keep an eye on COVID-19’s cardiovascular issues and identify the risk factors for a poor prognosis (e.g., age, smoking, obesity, blood pressure, etc.).

## 6. Viral and Bacterial Infections-Related Cardiotoxicity

It is now accepted that adverse cardiac events, such as myocardial infarction, arrhythmia, and heart failure, are a significant cause of death both during and after the hospitalization of older patients for pneumonia [49]. The most common cause of CAP and sepsis is streptococcus pneumoniae, and adult hospitalization is associated with a roughly 19% chance of a negative cardiac event (e.g., heart failure, arrhythmia, infarction) [71]. Cardiomyocyte contractility appears to be inhibited, in vitro, after exposure to purified *S. pneumoniae* cell wall and *S. pneumoniae* in the bloodstream is also capable of translocation into the myocardium. Within the ventricles, *S. pneumoniae* forms discrete bacteria-filled lesions approximately 10–100 μm in diameter (i.e., microlesions) [72].

Both experimentally infected nonhuman primates and two of nine human postmortem cardiac samples from people who died of invasive pneumococcal illness showed evidence of microlesion development [73]. The pyruvate oxidase enzyme, which is produced by pneumococcus, is also used to produce large amounts of hydrogen peroxide (H2O2) [74]. Cardiomyocytes nearby pneumococci within microlesions are probably exposed to large levels of this harmful reactive oxygen species, which is also capable of harming cell membranes, as catalase in the blood neutralizes H_2_O_2_ [75].

It is not clear whether systemic lung inflammation or direct infection of heart tissue causes the typical consequence of severe influenza virus infection known as cardiac dysfunction [76]. Myocardial dysfunction and damage were often noted during the most severe recent influenza pandemic, which occurred in 2009, and they were linked to increased mortality [77]. The goal of the study by Filgueiras-Rama et al. was to examine the human influenza virus’s potential to infect the murine myocardium and to determine the association between this capacity and molecular changes. In these investigations, two recombinant viruses were utilized, one with enhanced (PAmut) and one with attenuated pathogenicity (PB2mut). Following intranasal inoculation with the influenza A virus, viral titers were found in both the heart and the lungs, but they did not correlate with one another, indicating that the severity of heart infection is not correlated with the severity of lung infection. Viral mRNA (NEP) and proteins (NP) were found, which supported viral replication in the myocardium. Prior to the established weight drop of 75% of the initial body weight, infected mice showed higher mortality, raising concerns about possible arrhythmic premature death [78].

## 7. Limitations of Current Literature

Data regarding cardiovascular events in pneumonia originating mainly from retrospective studies or administrative database with several methodological limitations. Lack of adequate adjustment for confounders, comorbidities and acute alterations such as septic shock and electrolyte abnormalities may represent risk factors, as well relevant confounders in case cardiovascular events, are identified. Moreover, cardiovascular complications are usually not stratified by severity. Finally, few animal models have been developed to understand the association of these cardiovascular events with CAP.

## 8. Clinical Implications and Research Future Perspectives

Future efforts should focus on the clinical implications of cardiovascular events for patients with pneumonia, especially in the first 24 h after hospital admission. CAP individuals at risk for cardiac events may benefit from stratification since it allows for the development of diagnostic tools, monitoring systems, and preventative measures. The identification of patients with CAP who are at high risk for cardiac complications could assist clinicians in their decision-making regarding the disposition of patients admitted to the hospital for CAP and their discussions with these patients about their short- and long-term prognosis. All hospitalized patients should undergo EKG (e.g., ischemic changes, arrhythmias, QTc prolongation) with or without pro-BNP/troponin evaluation, while an extensive cardiac evaluation with echocardiography should be individualized based on risk factors. However, the best approach to diagnose and prevent CV events in patients admitted for CAP is still unknown and should be framed, for example, by the possibility of unnecessary testing in some patients and cost-effectiveness-related issues. For CAP patients who are still at a high risk of experiencing cardiovascular events throughout the course of a long-term follow-up, a strict follow-up after discharge should be planned. In order to lower the incidence of CAP in these high-risk populations, physicians and health officials must step up their efforts to maximize the rates of influenza and pneumococcal immunization among the elderly and those with chronic heart diseases. From a research perspective, a number of areas need more study. The pathophysiology of cardiovascular events during pneumonia is a challenging problem that is far from being solved. Some mechanisms, such as how pneumonia affects catecholamines, pulmonary circulation, renal function, fluid and sodium balance, and pro-coagulation state, need to be properly researched circulation, renal function and fluid and sodium balance, and pro-coagulation state [79]. Second, it would be beneficial to have a clearer understanding of the frequency of additional problems related to pneumonia and how they affect CAP patients’ outcomes. As a result, it is important to combine various worldwide databases in order to examine problems involving other organs, such as pulmonary embolism, delirium, or renal impairment, as well as AMI, stroke, arrhythmia, and heart failure. To determine if this model might be useful in investigations of clinical outcomes as well as preventive or therapeutic strategies, it is important to examine the physiopathological classification of plaque-related vs. plaque-unrelated events in patients with pneumonia.

## 9. Conclusions

Cardiovascular complications may occur in CAP patients during hospitalization as well as after discharge. With an increase in both short- and long-term mortality, these events have a significant impact on the outcomes for patients.

The underlying mechanisms of heart diseases concurrent with pneumonia are distinguishable as plaque-related, such as the AMI, or plaque-unrelated including arrhythmias and heart failure. There is an urgent need to systematically investigate novel diagnostic methods, risk stratification scales, and cardioprotective therapies given the persistently poor mortality outcomes in patients with CAP and the influence of cardiac events on outcomes.

## Figures and Tables

**Table 1 microorganisms-10-02177-t001:** Risk factors associated with cardiovascular events in CAP.

Category	Risk Factor
Host	Ageing Nursing home resident Male sex Smoking COPD Diabetes Alcohol abuse Heart disease Kidney disease Obesity Arterial hypertension
Microorganisms	Staphilococcus aureus Streptococcus pneumoniae Chlamydophila pneumoniae Mycoplasma pneumoniae Influenza virus
Clinical presentation	Tachypnoea ≥ 30 rpm Na < 130 mmol Haematocrit < 30% pH 7.35 BUN ≥ 30 mg·dL High BNP levels
Radiological presentation	Multilobar involvement Pleural effusion
Severity	Sepsis PSI CURB65
Antibiotic treatment	Macrolide Fluoroquinolones

## Data Availability

All data are included in the main text.

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
