# Peer review of "Cardiovascular Complications in Community-Acquired Pneumonia"

_microorganisms, 2022, doi:10.3390/microorganisms10112177_

Round 1

Reviewer 1 Report

I would like to congratulate the authors for their interesting and informative paper.

This is a narrative review of the literature on the cardiovascular complications observed in patients with community-acquired pneumonia. Specifically, the authors present and discuss the pathophysiological mechanisms, prevalence, risk factors, outcomes, and treatments of cardiovascular events in patients with community-acquired pneumonia. They conclude that diagnostic, monitoring, and preventive measures can be applied to patients at high risk for cardiac events following appropriate stratification, and they urge for prospective studies on cardioprotective treatments in this population.

This is an overall well-written review. The importance of the article is explicitly justified, and the specific aims of the review are thoroughly and clearly stated. Furthermore, statements that are essential for the arguments of the manuscript are supported by references. Finally, the relevant outcome data are presented with appropriate effect sizes and confidence intervals.

Here, I have made a few suggestions that (in my opinion) could help to further improve the quality of the manuscript.

·         The authors may consider describing the performed literature search. Although a detailed description (as in a systematic review) is not necessary, specification of search terms and types of included articles will significantly increase the transparency about the sources of information on which the paper is based.

·         The authors may consider mentioning the design and associated level of evidence of all referenced studies to increase the quality of the scientific points made in the manuscript.

·         The authors may consider discussing the large study by Corrales-Medina et al. on the cardiac complications occurring in patients with community-acquired pneumonia published in Circulation (DOI: 10.1161/CIRCULATIONAHA.111.040766).

·         The authors may consider further discussing tools for risk stratification of patients with community-acquired pneumonia that assess the risk of incident cardiac complications.

Author Response

Dear Dr. Editor,

The authors would like to thank the editor for giving them the opportunity to submit a revised version of the paper and the reviewers for their thoughtful analysis of the manuscript. We agree with all comments and recommendations suggested by the reviewers. We have changed the manuscript to comply with reviewers’ recommendations (all the changes are in red in the main text). The following is a detailed response to each reviewer’s recommendations.

Best wishes

Antonio

Reviewer 1:

I would like to congratulate the authors for their interesting and informative paper. This is a narrative review of the literature on the cardiovascular complications observed in patients with community-acquired pneumonia. Specifically, the authors present and discuss the pathophysiological mechanisms, prevalence, risk factors, outcomes, and treatments of cardiovascular events in patients with community-acquired pneumonia. They conclude that diagnostic, monitoring, and preventive measures can be applied to patients at high risk for cardiac events following appropriate stratification, and they urge for prospective studies on cardioprotective treatments in this population. This is an overall well-written review. The importance of the article is explicitly justified, and the specific aims of the review are thoroughly and clearly stated. Furthermore, statements that are essential for the arguments of the manuscript are supported by references. Finally, the relevant outcome data are presented with appropriate effect sizes and confidence intervals. Here, I have made a few suggestions that (in my opinion) could help to further improve the quality of the manuscript.

Response: We would like to thank the reviewer for her/his nice words.

Comment 1: The authors may consider describing the performed literature search. Although a detailed description (as in a systematic review) is not necessary, specification of search terms and types of included articles will significantly increase the transparency about the sources of information on which the paper is based.

Response to comment 1: We thank the reviewer for her/his comment. We recognize that this comment is important. Thus, we added a paragraph at the end of the introduction as follows “We searched Medline from inception to 1st August 2021 for articles in English evaluating the incidence of subsequent cardiovascular events during and after a CAP episode (community-acquired pneumonia” AND (“complications” OR “acute coronary syndromes” OR “heart failure” OR “arrhythmia” OR “atrial fibrillation” OR “stroke” OR “acute myocardial infarction”). We also retrieved previously published reviews on the subject and scanned their references for any additional missed publications.”

Comment 2: The authors may consider mentioning the design and associated level of evidence of all referenced studies to increase the quality of the scientific points made in the manuscript.

Response to comment 2: We thank the reviewer for her/his comment. We recognize that this comment is important. However, must of the studies are based on retrospective data and some of them are based on preclinical studies. For these reasons, we added a paragraph on limitations of current literature.

“ 7. Limitations of current literature

Data regarding cardiovascular events in pneumonia originating mainly from retrospective studies or administrative database with several methodological limitations. Lack of adequate adjustment for confounders Comorbidities and acute alterations such as septic shock and electrolyte abnormalities may represent risk factors as well relevant confounders in case cardiovascular events are identified. Moreover, cardiovascular complications are usually not stratified by severity. Finally, few animal models have been developed to understand the association of these cardiovascular events with CAP.”

Comment 3: The authors may consider discussing the large study by Corrales-Medina et al. on the cardiac complications occurring in patients with community-acquired pneumonia published in Circulation (DOI: 10.1161/CIRCULATIONAHA.111.040766).

Response to comment 3: We thank the reviewer for her/his comment. We recognize that this comment is important. The study conducted by Corrales-Medina et al represents a milestone in the interpretation of cardiovascular events and risk factor for them in CAP, in particular it was conducted as a prospective study. For this reason, we add a sentence in the text and we include the risk factors identified by the study in the Table.

Comment 4: The authors may consider further discussing tools for risk stratification of patients with community-acquired pneumonia that assess the risk of incident cardiac complications.

Response to comment 4: We thank the reviewer for her/his comment. We recognize that this comment is important. Thus, we add a table to clarify and categorize risk factors. Furthermore, we modify the text to discuss the importance of risk stratification in CAP as follows: “Clinical risk factors are categorized in Table 1.

Identifying the most vulnerable patients who are likely to develop cardiovascular events can be challenging. For both therapeutic and scientific objectives, clinical methods that categorize CAP patients thought prediction scores based on their propensity to experience cardiac problems may be helpful. Using clinical and laboratory data available at the time of hospital admission, recent experiments have attempted to develop a prediction score to stratify the short-term risk of cardiac complications. [10,48,56,57]

In 2013, Viasus and colleagues conducted a prospective monocenter study to identify a prediction score which stratifies low-, intermediate- and high-risk groups [10,48]. Cor-rales-Medina et al. developed and validated a prediction rule for the short-term risk of cardiac complications with more accuracy than the PSI score, using standard clinical and laboratory data. this strategy may be useful to validate a timely risk stratification in clinical and research settings [10]. A major challenge in future CAP research will be the evaluation of cardiovascular biomarkers in addition to risk factors to improve clinical scoring and, thus, better identify patients at risk of cardiovascular events.”

Reviewer 2 Report

The authors present an updated review on the incidence of cardiac (CV) events after community-acquired pneumonia (CAP). The manuscript is well organized and provides the reader with a good overview on the subject.

The importance of this research topic is clearly discernible throughout the manuscript and as such confirms its interest to the journal’s readership. In this regard, there has been a growing number of reviews and meta-analysis already published, which provide relevant information on the estimate of CV events after CAP and should be cited in the manuscript (Tralhão A, Póvoa P. J Clin Med. 2020 Feb 3;9(2):414). Another excellent review worth citing is the one by Musher D et al N Engl J Med 2019; 380:171-176.

Furthermore, I have the following comments and questions:

- Line 38: there is a discrepancy between the different mortality timepoints and the mortality figures provided, i.e., there are 4 time points and only 3 figures.

- Line 42: the term cardiovascular sequelae should be modified, as it implies a long lasting deleterious effect. Acute coronary syndrome are, by definition, acute events.

- Line 44: it should be emphasized that CAP “may lead” instead of “causes” SIRS, especially if severe.

- Line 53: antithrombin is an antithrombotic mediator.

- Line 55: I don’t understand how the process of atherosclerosis is unrelated to a plaque-independent mechanism.

- Line 64 to 76: a distinction between the mechanisms of type 1 and type 2 myocardial infarction needs to be made.

- Line 113: a sentence on how acute renal impairment contributes for acute heart failure should be added.

- Lines 130 to 135: this paragraph is somewhat confusing. The first sentence seems to introduce the common pathophysiologic pathways between arrhythmias and AMI but then the epidemiology and order of events such as heart failure are mentioned. Please clarify.

- Lines 170-175: please provide some detail on the variables used to build the risk scores.

- Lines 184: “Statins are beneficial” should be substituted for “Statins may be beneficial”.

- Line 190: please add the p-value of the study [ref. 60].

- Line 229: please remove “It is a significant symptom of coronary heart disease”.

- Line 272: in the section “7. Clinical implications and research future perspectives”, it should be stressed that the best approach to diagnose and prevent CV events in patients admitted for CAP is still unknown and should be framed, for example, by the possibility of unnecessary testing in some patients and cost-effectiveness related issues.

Author Response

Dear Dr. Editor,

The authors would like to thank the editor for giving them the opportunity to submit a revised version of the paper and the reviewers for their thoughtful analysis of the manuscript. We agree with all comments and recommendations suggested by the reviewers. We have changed the manuscript to comply with reviewers’ recommendations (all the changes are in red in the main text). The following is a detailed response to each reviewer’s recommendations.

Best wishes

Antonio

Reviewer 2:

The authors present an updated review on the incidence of cardiac (CV) events after community-acquired pneumonia (CAP). The manuscript is well organized and provides the reader with a good overview on the subject. The importance of this research topic is clearly discernible throughout the manuscript and as such confirms its interest to the journal’s readership.

Response: We would like to thank the reviewer for her/his nice words.

Comment 1: In this regard, there has been a growing number of reviews and meta-analysis already published, which provide relevant information on the estimate of CV events after CAP and should be cited in the manuscript (Tralhão A, Póvoa P. J Clin Med. 2020 Feb 3;9(2):414). Another excellent review worth citing is the one by Musher D et al N Engl J Med 2019; 380:171-176.

Response to comment 1: We thank the reviewer for her/his comment. We recognize that this comment is important. According to the other reviewer comments we decided to improve methodology and we specified that reviews on the subject were scanned in order to find other relevant papers. Thus, we added a paragraph at the end of the introduction as follows “We searched Medline from inception to 1st August 2021 for articles in English evaluating the incidence of subsequent cardiovascular events during and after a CAP episode (community-acquired pneumonia” AND (“complications” OR “acute coronary syndromes” OR “heart failure” OR “arrhythmia” OR “atrial fibrillation” OR “stroke” OR “acute myocardial infarction”). We also retrieved previously published reviews on the subject and scanned their references for any additional missed publications.”

 Comment 2: - Line 38: there is a discrepancy between the different mortality timepoints and the mortality figures provided, i.e., there are 4 time points and only 3 figures.

Response to comment 2: We thank the reviewer for her/his comment. We recognize that this comment is important. Thus, we modify the sentence as follows “ Mortality for CAP is 3%, 23%, and 31%, at hospitalization, 30 days, and 1 year, respectively [3-6]. 

Comment 3: - Line 42: the term cardiovascular sequelae should be modified, as it implies a long lasting deleterious effect. Acute coronary syndrome are, by definition, acute events.

Response to comment 3: We thank the reviewer for her/his comment. We recognize that this comment is important. Thus, we modify the term cardiovascular sequelae in cardiovascular events.

Comment 4: - Line 44: it should be emphasized that CAP “may lead” instead of “causes” SIRS, especially if severe.

Response to comment 4: We thank the reviewer for her/his comment. We recognize that this comment is important. Thus, we modify the sentence according to the reviewer suggestion.

Comment 5: - Line 53: antithrombin is an antithrombotic mediator.

 Response to comment 5: We thank the reviewer for her/his comment. We recognize that this comment is important. Thus, we modify the sentence according to the reviewer suggestion.

Comment 6: - Line 55: I don’t understand how the process of atherosclerosis is unrelated to a plaque-independent mechanism.

 Response to comment 6: We thank the reviewer for her/his comment. We recognize that the sentence is confusing. Thus, we modify the sentence as follows “The pathophysiology of CAP-related cardiovascular complications results from the process of atherosclerosis leading to endothelial injury and myocardial ischemia or dysfunction.[24,25]”

Comment 7: - Line 64 to 76: a distinction between the mechanisms of type 1 and type 2 myocardial infarction needs to be made.

 Response to comment 7: We thank the reviewer for her/his comment. We recognize that this comment is relevant. Thus, we modify the paragraph as follows “Atherosclerosis and endothelial dysfunction are responsible for plaque rupture, thrombus creation, mismatch between arterial perfusion and myocardial oxygen request, and are the basis of different vascular pathologies [26]. Indeed, myocardial infarction occurs in those with atherosclerotic plaque rupture and thrombosis (type I), or in case of myocardial oxygen supply and demand imbalance in the context of an acute illness (type II). After plaque disruption, local cardiac ischemia may occur. The rupture of fibrous cap overlying a previously stable plaque is the primum movens of cardiovascular events, such as acute myocardial infarction (AMI) and unstable angina. Plaque rupture exposes the underlying lipid-rich core which is highly pro-thrombotic with the subsequent formation of a thrombus in situ and the occlusion of the vessel [27]. Platelet activation and aggregation on the surface of a ruptured plaque may be stimulated by pneumonia either directly through pro-inflammatory cytokines or by bacterial products [28,29]. All these processes, comprising the plaque rupture, in situ thrombus formation, and the balance of arterial perfusion versus myocardial oxygen demand may be triggered or worsened during an episode of pneumonia and, thus, lead to the occurrence of plaque-related cardiovascular events. ”

Comment 8: - Line 113: a sentence on how acute renal impairment contributes for acute heart failure should be added.

Response to comment 8: We thank the reviewer for her/his comment. We recognize that this comment is relevant. Thus, we modify the sentence as follows “Moreover, acute renal impairment is frequent in hospitalized patients with CAP and plays a pivotal role in heart failure by several mechanisms such as alteration in the renin-angiotensin-aldosterone system (RAAS), dysregulated pressure-sensing baroreceptors and cellular signaling, abnormal sympathetic nervous system mechanisms. [40,41].”

Comment 9: - Lines 130 to 135: this paragraph is somewhat confusing. The first sentence seems to introduce the common pathophysiologic pathways between arrhythmias and AMI but then the epidemiology and order of events such as heart failure are mentioned. Please clarify.

Response to Comment 9: We thank the reviewer for her/his comment. We recognize that this paragraph is confusing. Thus, we delete the paragraph.

Comment 10: - Lines 170-175: please provide some detail on the variables used to build the risk scores.

Response to comment 10: We thank the reviewer for her/his comment. We recognize that this comment is important and we decided to add a table on risk factors that are used to identify patients at risk of cardiovascular events in CAP. Moreover, we modify the text to discuss the importance of risk stratification in CAP as follows: “Clinical risk factors are categorized in Table 1.

Identifying the most vulnerable patients who are likely to develop cardiovascular events can be challenging. For both therapeutic and scientific objectives, clinical methods that categorize CAP patients thought prediction scores based on their propensity to experience cardiac problems may be helpful. Using clinical and laboratory data available at the time of hospital admission, recent experiments have attempted to develop a prediction score to stratify the short-term risk of cardiac complications. [10,48,56,57]

In 2013, Viasus and colleagues conducted a prospective monocenter study to identify a prediction score which stratifies low-, intermediate- and high-risk groups [10,48]. Cor-rales-Medina et al. developed and validated a prediction rule for the short-term risk of cardiac complications with more accuracy than the PSI score, using standard clinical and laboratory data. this strategy may be useful to validate a timely risk stratification in clinical and research settings [10]. A major challenge in future CAP research will be the evaluation of cardiovascular biomarkers in addition to risk factors to improve clinical scoring and, thus, better identify patients at risk of cardiovascular events.”

Comment 11: - Lines 184: “Statins are beneficial” should be substituted for “Statins may be beneficial”.

 Response to comment 11: We thank the reviewer for her/his comment. We modify the sentence according to reviewer suggestion.

Comment 12: - Line 190: please add the p-value of the study [ref. 60].

 Response to comment 12: We thank the reviewer for her/his comment. We add the p-value according to reviewer suggestion.

Comment 13: - Line 229: please remove “It is a significant symptom of coronary heart disease”.

 Response to comment 13: We thank the reviewer for her/his comment. We remove the sentence according to reviewer’s suggestion.

Comment 14: - Line 272: in the section “7. Clinical implications and research future perspectives”, it should be stressed that the best approach to diagnose and prevent CV events in patients admitted for CAP is still unknown and should be framed, for example, by the possibility of unnecessary testing in some patients and cost-effectiveness related issues.

 Response to comment 14: We thank the reviewer for her/his comment. We modify the sentence according to reviewer suggestion as follows “ Future efforts should focus on the clinical implications of cardiovascular events for patients with pneumonia, especially in the first 24 hours after hospital admission. CAP individuals at risk for cardiac events may benefit from stratification since it will allow for the development of diagnostic tools, monitoring systems, and preventative measures. Identification of patients with CAP who are at high risk for cardiac complications could assist clinicians in their decision-making regarding the disposition of patients admitted to the hospital for CAP and their discussions with these patients about their short- and long-term prognosis. All hospitalized patients should undergo EKG (e.g: ischemic changes, arrhythmias, QTc prolongation) with or without pro-BNP/troponin evaluation, while an extensive cardiac evaluation with echocardiography should be individualized based on risk factors. However, the best approach to diagnose and prevent CV events in patients admitted for CAP is still unknown and should be framed, for example, by the possibility of unnecessary testing in some patients and cost-effectiveness related issues.”

Round 2

Reviewer 1 Report

Thank you very much for considering my suggestions and comments and revising your manuscript accordingly.